# Hypersensitivity Pneumonitis: A Pictorial Review Based on the New ATS/JRS/ALAT Clinical Practice Guideline for Radiologists and Pulmonologists

**DOI:** 10.3390/diagnostics12112874

**Published:** 2022-11-20

**Authors:** Mona Dabiri, Maham Jehangir, Pegah Khoshpouri, Hamid Chalian

**Affiliations:** 1Department of Radiology, Children’s Medical Center, Tehran University of Medical Science, Tehran 14176-14411, Iran; 2Cardiothoracic Imaging, Department of Radiology, Rush University Medical Center, Chicago, IL 60612, USA; 3Department of Radiology, University of Washington, Seattle, WA 98105, USA; 4Cardiothoracic Imaging, Department of Radiology, University of Washington, Seattle, WA 98105, USA

**Keywords:** hypersensitivity pneumonitis, thoracic imaging, fibrotic HP, non-fibrotic HP

## Abstract

Hypersensitivity pneumonitis (HP) is a complicated and heterogeneous interstitial lung disease (ILD) caused by an excessive immune response to an inhaled antigen in susceptible individuals. Accurate diagnosis of HP is difficult and necessitates a detailed exposure history, as well as a multidisciplinary discussion of clinical, histopathologic, and radiologic data. We provide a pictorial review based on the latest American Thoracic Society (ATS)/Japanese Respiratory Society (JRS)/Asociación Latinoamericana del Tórax (ALAT) guidelines for diagnosing HP through demonstrating new radiologic terms, features, and a new classification of HP which will benefit radiologists and pulmonologists.

## 1. Introduction

Exposure to inhaled antigens is considered the main cause of hypersensitivity pneumonitis (HP), a complex form of interstitial lung disease (ILD) [1,2]. As part of the initial diagnostic evaluation, a high-resolution computed tomography (HRCT) scan of the chest is commonly performed. In an appropriate clinical setting, a certain HRCT pattern can be suggestive of HP [3]. Individuals who do not have a definite ILD diagnosis following clinical and HRCT screening are frequently submitted to more invasive diagnostic tests (e.g., bronchoscopy or surgical lung biopsy) to get a diagnosis. HRCT patterns linked to HP have been studied in several investigations. A radiologist’s confident diagnosis of hypersensitivity pneumonitis is correct 88–92% of the time [4,5].

In this pictorial review we aim to familiarize readers with the new nomenclature, redefine classification of HP and associated radiological findings based on the recent American Thoracic Society (ATS)/Japanese Respiratory Society (JRS)/Asociación Latinoamericana del Tórax (ALAT) clinical practice guidelines.

## 2. Epidemiology

The estimated incidence of HP is 1.3 to 1.9 per 100,000 people per year in the United States general population [6]. After idiopathic pulmonary fibrosis (IPF) and connective tissue disease-related interstitial lung disease, HP is the third most frequent ILD. More than 200 sensitizing antigens have already been identified and these can be present in the environment, at work, at home, and during leisure activities. Fungi, bacteria, probiotics, low-molecular-weight chemical compounds, protozoa, and animal proteins are the major antigen sources [7].

HP has a predilection for women and older individuals of 65 years of age or above, with the average patient receiving a diagnosis in their fifth or sixth decade. However, it can also be found in children and younger adults [3].

## 3. Definition, Classification and Clinical Features of HP

HP, also previously known as extrinsic allergic alveolitis [8], is the culmination of a dysregulated immune response to a provocative inhaled antigen appearing as inflammation and/or fibrosis of the lung parenchyma and small airways [3]. The diversity of clinical manifestations and the progression of HP heavily depends on the nature of the causal agent (occupational or environmental agents such as fungal, bacterial, and avian) [9], the duration of exposure, and host factors [10]. Cough and dyspnea are common symptoms. Occasionally, patients may present with fever, weight loss, and fatigue. Physical examination may reveal crackles and inspiratory squeaks and digital clubbing can be seen with advanced disease [8].

HP was previously classified as acute, subacute, or chronic depending on the duration of symptoms [11], which is no longer considered clinically useful. Fibrosis is an important predictor of prognosis [12]; therefore, the most recent guidelines for the diagnosis of HP have classified it as non-fibrotic (purely inflammatory) HP or fibrotic (mixed inflammatory and fibrotic or purely fibrotic) HP [3]. Fibrotic HP (FHP) is more severe and has a worse prognosis, 28% with 4-year mortality and 52% with 7-year mortality [13], as opposed to non-fibrotic HP (NFHP) which may be recurrent [14]. Patients with "cryptogenic HP" or “HP of unclear cause” are those in whom a culprit exposure has not been discovered but who otherwise have HP-like symptoms. Table 1 highlights some important demographic and clinical features of HP [3].

## 4. Pathogenesis of HP

Following inhalation of an inciting antigen or a combination of antigens (Table 2), a susceptible host generates an excessive immunological response [15,16]. HP susceptibility has been linked to several genetic variations, including those in the major histocompatibility complex class II [17,18]. Inciting antigens interact with host proteins to create haptens [19]. Viruses can cause or aggravate hypersensitivity to environmental antigens by increasing the antigen-presenting capacity of alveolar macrophages, decreasing antigen clearance, and increasing the production of inflammatory cytokines [20]. Tobacco smoking augments immune responses to inciting antigens, accelerating the pathogenetic process that leads to fibrosis [21,22]. According to the latest research, less than 40% of patients’ causative antigens can be recognized [13]. Identifying antigens and exposure sources in some cases remains unattainable due to reasons such as; incomplete environmental history and information on patient’s recent exposures and lack of information about the relationship between specific antigen and HP development. Confirming inciting antigens that cause HP can be achieved in the context of casual inference: (A) strength, (B) consistency, (C) dose–response, (D) temporality, (E) reversibility, (F) biological plausibility, (G) specificity, and (H) analogy [23]. 

Both humoral and cellular processes have a role in HP (Figure 1). The inflammatory response is primarily mediated by T-helper cells and antigen-specific immunoglobulin (Ig)G antibodies after antigen exposure and processing by the innate immune system, resulting in the accumulation of lymphocytes and the formation of granulomas [17]. While the pathogenesis of pulmonary fibrosis is unknown, it is thought that in fibrotic diseases, abnormal repair mechanisms in response to recurrent alveolar epithelial injury result in fibroblast activation and proliferation, extracellular matrix accumulation, and eventual destruction of the lung architecture [24].

**Table 2 diagnostics-12-02874-t002:** Exposures and causal antigens [10,25].

Antigen	Source
*Trichosporon* spp.	Damo wooden hoses (Japan)
Argan	Cosmetic factory, hair salons
Chinchilla	Pet chinchilla
*Aureobasidium pullulans*	Domestic fungal contamination
*Rhizopus* spp.	Sawmill worker
*Mucor* spp.	Moldy wood
Beryllium, Cobalt, Zinc	Batteries, hard metal alloys, zinc fumes
*Thermoactinomyces* spp.	Farm environment, domestic bacterial contamination, garbage exposure
*Saccharopolyspora rectivirgula*	Farm environment, esparto grass
*Nontuberculous mycobacterium*	Hot tub
*Wallemia sebi*	Farm environment
*Pseudomonas* spp.	Cork factory, home humidifier
Protein of bloom, droppings, feather	Chickens, Budgerigars, Pigeons, Cockatiels
*Aspergillus* spp., *Penicillium* spp., *Cladosporium* spp.	Mold dust
Fungi and molds	Contaminated water
*Acinetobacter* spp.	Contaminated machine fluid
Achromobacter	Contaminated humidifiers
*Bacillus* spp.	Contaminated water, sawduct, moist
*Streptomyces albus*	Contaminated compost, mushroom, hay dusty soil
*Methyl acrylates*	Dental technicians
Isocyanate acid anhydrides	Plastic, paint, glue, varnish, resins
*Aspergillus fumigatus*, *Thermophilic actinomycetes*	Organic waste, soil

## 5. Imaging of HP

### 5.1. Chest HRCT Scanning Protocol

The suggested scanning protocol for suspected HP examination is provided in the ATS/ERS/JRS/ALAT diagnosis-of-IPF guidelines [26]. It is based on high-resolution scanning of the chest, with particular care paid to parameter selection to ensure motion-free images and appropriate image quality at a low radiation dosage. Three sets of images are usually obtained for HRCT evaluation of ILD: (A) Axial scan in supine position after deep inspiration, for which helical (volumetric CT) acquisition is recommended. (B) Axial scan in supine position after prolonged expiration, for which one can employ volumetric scanning or incremental scanning. (C) Axial scan in prone position after deep inspiration which can also be obtained via volumetric or incremental scanning.

Incremental scanning in the prone position and expiratory phase imaging should be obtained with non-irradiated increments of 10–20 mm or more to decrease the radiation dose. Nominal slice thickness for axial and helical CT should be 1.5 mm or less. Gantry rotation speed of 1 s or less is acceptable. In volumetric acquisition, highest pitch, shortest rotation time, and sub-millimetric collimation should be used. Tube potential and tube current should be according to patient’s size, which is typically 120 kVp (kilovolt peak) and 240 mAs (milliampere-seconds) and/or less. For patients with a lower body size, lower tube potential (100 kVp) is recommended. Tube current modulation is used to decrease radiation exposure [3,27]. In order to solve image noise, iterative reconstruction techniques (with special caution) should be utilized [28]. Utilizing the high-spatial-frequency reconstruction algorithm (e.g., bone algorithm) is recommended. However, using a sharp reconstruction algorithm should be inhibited as it can create image noise [29]. Except for air trapping, which is an expiratory HRCT finding, all signs of lung infiltration can be seen on inspiratory images [3]. For accurate lung parenchyma and small airway evaluation, intravenous iodinated contrast should not be administered [30].

### 5.2. HRCT for Diagnosis of HP

There is a large spectrum of sensitivity and specificity in both BAL, tissue sampling, and serologic findings in HP. In addition, obtaining a thorough history of exposures to related antigens is difficult [31]. Therefore, this highlights the significance of HRCT for radiologists and pulmonologists, so that they can recognize fibrotic and non-fibrotic HP. HRCT is preferable due to its sensitivity to detect and evaluate lung abnormalities through suggesting radiographic patterns and distinguishing fibrosis [30]. Typical HP, Compatible with HP, and Indeterminate for HP are the classification schemes proposed by the ATS/JRS/ALAT Clinical Practice guideline for HRCT patterns related with NFHP and FHP [3].

The usual computed tomography (CT) presentations of HP are similar to the bronchiolocentric inflammation seen in histology, which results in small, ill-defined ground-glass nodules that are widely distributed throughout all lung zones. This bronchiolocentric inflammation may also cause minor airway constriction, resulting in lobular air-trapping. Ground-glass opacities and an increase in lung parenchyma density may result from more extensive interstitial inflammation, with vessels and bronchial walls remaining visible. This ground glass opacity usually has a patchy distribution in HP, which is referred to as mosaic attenuation [1] (Figure 2).

According to the latest classification (according to the New ATS/JRS/ALAT Clinical Practice Guideline), HP is divided into fibrotic and non-fibrotic forms: NFHP is a mild form of HP compared to FHP. However, NFHP often progresses to fibrosis, yet is recurrent [32]. Diffuse infiltrative parenchymal abnormalities with GGO or mosaic attenuation, as well as at least one abnormality suggesting small airway disease (ill-defined, 5 mm centrilobular nodules on inspiratory images or air trapping on expiratory images) are important HRCT characteristics of Typical NFHP. Mosaic attenuation in NFHP is typically caused by lobules with pneumonitis (high attenuation) juxtaposed with lobules with normal or lower attenuation due to bronchial congestion. Features of uniform or subtle GGO, airspace consolidation, or lung cysts which are 3 to 25 mm [31], may be regarded as Compatible with NFHP in the appropriate clinical context [11]. These patterns are often bilateral and symmetric and diffuse both in the axial and craniocaudal views of the lungs.

Concomitant fibrosis and bronchiolar obstruction indicate FHP. Lung fibrosis is more common in the mid/lower lung zones in Typical FHP, or it is evenly distributed with basilar sparing and no central or peripheral predominance. Ill-defined centrilobular nodules, GGO, mosaic attenuation, air trapping, and/or the three-density pattern, which is highly specific for FHP, are all signs of bronchial obstruction [33]. HRCT features of Compatible with FHP show fibrosis pattern (UIP or extensive GGO with superimposed fibrosis) and distribution (upper lobe, peri-broncho-vascular, or subpleural predominance) of small airway disease, which is represented by ill-defined centrilobular nodules, or three-density pattern and/or air trapping on HRCT. Fibrosis without coexisting bronchial obstruction (UIP, nonspecific interstitial pneumonitis (NSIP), or organized pneumonia pattern) is indeterminate for FHP [11]. In patients with fibrotic HP without occupational exposure or history of smoking, emphysema is seen in more than 15% on chest radiographs, and 27% on HRCT [33].

Extensive lobular air trapping, centrilobular nodules, and the absence of a lower zone predominance to fibrosis can all assist in ruling out idiopathic pulmonary fibrosis (IPF) as a viable diagnosis [34].

### 5.3. Radiologic Terms for Heterogeneous Lung Attenuation

There are several radiologic terms linked with heterogeneous lung attenuation that need to be known and differentiated by radiologists to ensure correct final diagnosis in suspected HP cases. These include mosaic attenuation, air trapping, mosaic perfusion, and three-density pattern.

“Mosaic attenuation” is a term reserved for inspiratory phase CT. It is defined as sharply demarcated areas of low and high attenuation. Mosaic attenuation can be seen in vascular diseases, airway diseases, or infiltrative diseases [35] (Figure 2 and Figure 3). As opposed to mosaic attenuation, the term “air trapping” is only used on expiratory phase CT (Figure 2, Figure 3 and Figure 4). This term represents abnormal air retention distal to airway obstruction and manifests as areas of lucent lung on a background of normal relatively high attenuation lung parenchyma on expiratory imaging [3]. Like mosaic attenuation, the term “mosaic perfusion” (Figure 5) is also reserved for inspiratory CT. In mosaic perfusion abnormality, one may see a smaller caliber of the pulmonary vasculature within areas of low attenuation compared to normal vessel caliber in areas of higher attenuation. However, mosaic perfusion can be seen because of a vascular disease (pure perfusion abnormality), or in airway diseases where there is mosaic perfusion secondary to abnormal ventilation. Hence, the role of expiratory CT to aid differentiation of mosaic perfusion is secondary to vascular diseases from airway disease. If there is a similar gradient of attenuation between low and high attenuation areas on inspiratory and expiratory scans, mosaic perfusion is secondary to a vascular disease. If a higher attenuation gradient is measured on expiratory scans, an airway disease is the likely cause of mosaic perfusion [36].

Given the fact that most people do not relate to the “headcheese” sign [37], the term “three-density pattern” (Figure 6 and Figure 7) is now a favored terminology. This term is used when there is both an obstructive and infiltrative process in addition to areas of normal lung parenchyma manifesting as clearly delineated zones with three different attenuations [38]. The manifestations of obstructive abnormalities are areas of decreased attenuation and decreased vascularity with no significant increase in attenuation on expiratory scans. Areas with an infiltrative disorder manifest as GGO or even consolidative opacities with increase in attenuation on expiratory scans. Areas of interposed normal lung present as demarcated normal lungs with expected increased attenuation on expiratory scans. Three-density pattern is specific for fibrotic HP [3,39].

### 5.4. HRCT Patterns of HP

Three categories of NFHP (Table 3, Figure 8) and FHP (Table 4, Figure 9, Figure 10, Figure 11 and Figure 12) have been described.

## 6. Histopathologic Features of HP

There are three histopathological categories for FHP and NFHP: typical, probable, and indeterminate, details of which are beyond the scope of this paper. Notably, absence of these four features in any biopsy site improves the histopathological diagnostic confidence for HP: plasma cells more than lymphocytes, lymphoid hyperplasia, extensive well-formed sarcoidal with or without necrotizing granulomas, and aspirated particles. Presence of these features is suggestive of an alternate diagnosis (Table 5).

For typical histopathologic diagnosis of NFHP, at least one biopsy site should include: cellular interstitial pneumonia, cellular bronchiolitis, poorly formed non-necrotizing granulomas, and lack of characteristics which are mentioned for histopathological diagnostic confidence suggesting alternative diagnosis. Probable NFHP includes the criteria that are the same as for Typical HP, except poorly formed non-necrotizing granulomas in at least one biopsy site. Indeterminate for HP of NFHP is characterized by cellular interstitial pneumonia and cellular bronchiolitis and idiopathic interstitial pneumonitis patterns (cellular non-specific interstitial pneumonia, organizing pneumonia pattern, and peribronchiolar metaplasia without other features to suggest fibrotic HP), as well as absence of the features mentioned above.

Typical histopathological features of FHP comprise of chronic fibrosing interstitial pneumonia, airway centered fibrosis, and poorly developed non-necrotizing granulomas in at least one biopsy site, as well as absence of features in any biopsy site to suggest an alternative diagnosis which are mentioned above. Probable FHP, similar to Typical FHP, shares many of the same characteristics as FHP except the granulomas. In Indeterminate for HP of FHP, chronic fibrosing interstitial pneumonia should exist in at least one biopsy site as well as absence of features in any biopsy site to suggest an alternative diagnosis. The distinguishing feature for FHP depends on identification of centriacinar fibrotic lesions and features of NFHP as well as airway-centered fibrosis and parabrachial metaplasia. These findings can be a great help in differing HP from other fibrotic lung disorders [3,36].

## 7. Diagnostic Algorithm and Challenging Scenarios

Multidisciplinary discussion is helpful for making the correct diagnosis but identifying the culprit exposure is not sufficient, as up to half of patients with FHP do not have any identifiable exposure [40].

Patients with NFHP have an abrupt onset of symptoms which can be identified with centrilobular nodularity observed on HRCT imaging and BAL lymphocytosis. Patients with FHP, on the other hand, have an onset of symptoms gradually over time, as well as a nonspecific BAL cell accompanied with fibrotic features in HRCT [36].

The new guideline emphasizes three domains: Firstly, exposure identification, which includes clinical history, antigen-specific serum IgG with or without inhalational challenge. Secondly, radiologic findings. Finally, BAL lymphocytosis with or without histopathologic findings. A threshold of >30% lymphocytosis has been deemed a reasonable threshold as per the ATS guidelines. According to the level of diagnostic confidence, there are four levels: definite (90% or more), high confidence (80–89%), moderate confidence (70–79%), and low confidence (less than 69%). For levels 2, 3 and 4, multidisciplinary discussion with or without subsequent histopathological sampling is needed (Table 6) [3].

Air trapping is a non-specific finding which reflects small airway disease that can be found as non-HP ILDs [41], including IPF, sarcoidosis, and connective tissue disease-associated ILD, such as RA-ILD [42]. For instance, mosaic attenuation and air trapping in the setting of RA is a manifestation of constrictive bronchiolitis and is more common in females, those with positive RF (rheumatoid factor), and longstanding untreated disease. In addition, it can also be due to toxicity with certain medications including gold, penicillamine, rituximab, and sulfasalazine [43]. This highlights the need for relevant serological testing and a review of patient history to explore non-HP related causes of air trapping. However, there is no definitive gold-standard test to differentiate FHP from IPF, wherein overlapping imaging and histological findings often pose a diagnostic challenge. Notably, several studies have shown that presence of mosaic attenuation or air trapping itself does not exclude a diagnosis of IPF, however air trapping in the upper lobes is rarer in IPF compared to FHP [4,44,45]. Barnett and colleagues found that a threshold of five or more lobules of mosaic attenuation in each of three or more lobes bilaterally improved the diagnostic specificity for FHP, however no threshold for mosaic attenuation could completely exclude a diagnosis of IPF. In contrast to mosaic attenuation, the three-density pattern (previously referred to as headcheese sign) was found to be highly specific for a high-confidence diagnosis of FHP as an alternate diagnosis to IPF [46]. Another distinguishing feature of FHP from IPF is upper lung predominant fibrosis [45]; however, only about 10% of patients with FHP have upper lung predominant disease. As per the recent ERS/ATS guidelines, a UIP pattern of fibrosis with significant concomitant air trapping may qualify as a pattern “Compatible with FHP” but does not meet criteria for “Typical FHP”. Therefore, in such cases a final diagnosis of HP should only be made using a comprehensive multidisciplinary approach wherein all imaging patterns should be interpreted in the context of other key features including exposure identification, BAL lymphocytosis, and, if needed, histological diagnosis. Other helpful indicators that increase the likelihood of an HP diagnosis include female sex, absence of smoking history, mid-inspiratory squeaks on auscultation, and an obstructive or mixed restrictive/obstructive physiology. Identification of a clear exposure preceding symptom onset is more frequent in NFHP and up to 50% of patients with FHP may not have an identifiable exposure. In view of these challenges, the guidelines have emphasized the role of clinicians in taking a thorough history to identify potential exposures relevant to the patient’s occupation, geographical location, and cultural habits. It has been suggested that serum IgG testing against potential antigens associated with HP may help to explore the source of the potential causative agent, particularly when other diagnostic imaging, BAL lymphocytosis, and biopsy findings of HP are less certain.

## 8. Summary

In order to identify HP from other ILDs, clinicians are encouraged to take a thorough history of environmental exposures to antigens (using serum IgG) and the onset of symptoms. Suspected patients may also undergo BAL lymphocytosis. These approaches should be combined with findings on HRCT. Final diagnosis should be discussed and concluded after multidisciplinary discussions. This pictorial review provides radiologists and pulmonologists with a valuable guide to diagnosis of HP based on the new ATS/JRS/ALAT Clinical Practice Guidelines.

## Figures and Tables

**Figure 1 diagnostics-12-02874-f001:**
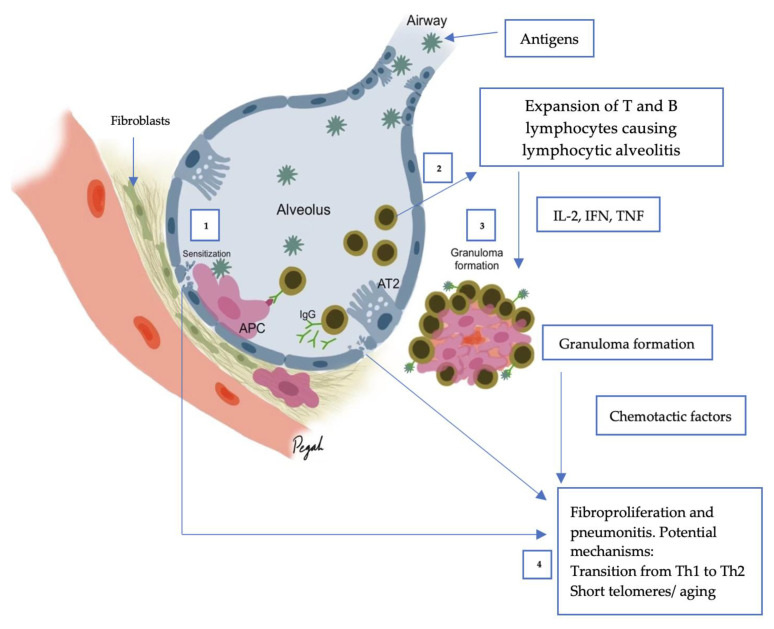
Immunopathogenesis of HP. Inhaled antigens interact with antigen-presenting cells (APCs, i.e., macrophages, dendritic cells) via pattern recognition receptors including toll-like receptors (1). APCs stimulate a T-helper 1 cell (Th1) response. Neutrophils are present in early disease. Stimulated plasma cells (B cells) (2) produce IgG antibodies (humoral response) which triggers the complement cascade (3) and enhances macrophages which fuse to multinucleated giant cells and epithelioid cells to form granulomas, mediated by Th1 cytokines. Granulomas produce chemotactic factors which, in combination with dysregulated T cell function, promotes fibroblast proliferation (4). Fibroblasts differentiate into myofibroblasts, produce collagen and extracellular matrix, causing fibrosis.

**Figure 2 diagnostics-12-02874-f002:**
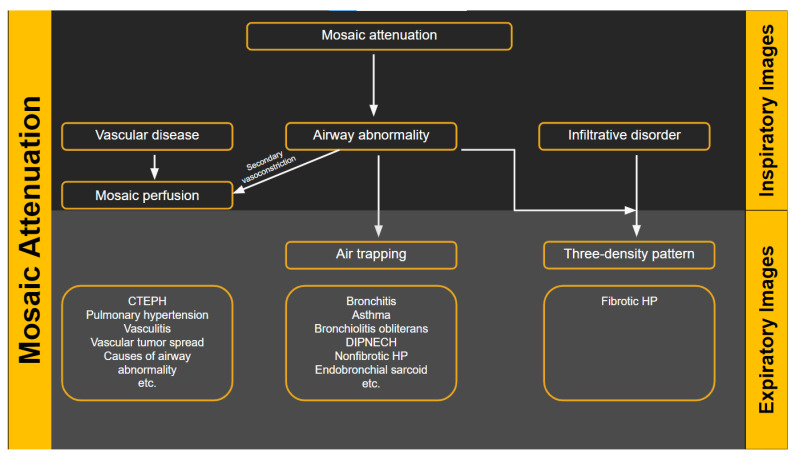
Flow diagram showing the heterogeneous lung attenuation patterns. Mosaic attenuation is a terminology reserved for inspiratory phase imaging and can be seen in vascular diseases, small airway disease or infiltrative diseases such as hypersensitivity pneumonitis. Mosaic perfusion is a feature of primary vascular disease but can also be seen with small airway disease due to hypoxic vasoconstriction.

**Figure 3 diagnostics-12-02874-f003:**
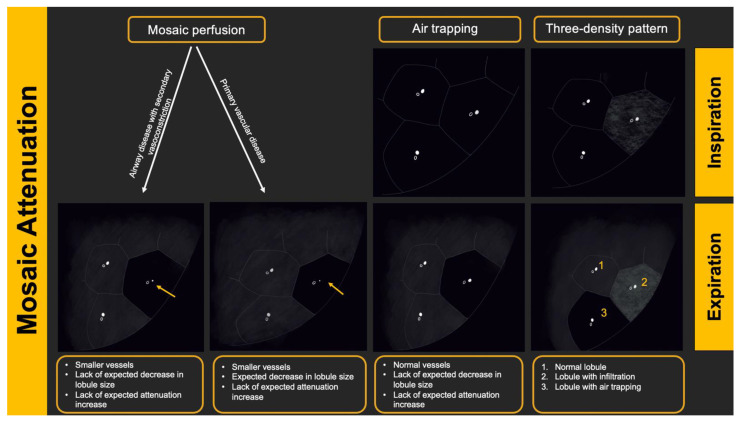
Pictorial representation of heterogeneous lung attenuation patterns at the level of the secondary pulmonary lobule. Top panel shows expected appearance on inspiratory images and the bottom panel shows changes in lung parenchymal density on expiratory images. Notice that mosaic perfusion can be seen with primary vascular disease and small airway disease, with change in lobule size being the differentiating feature.

**Figure 4 diagnostics-12-02874-f004:**
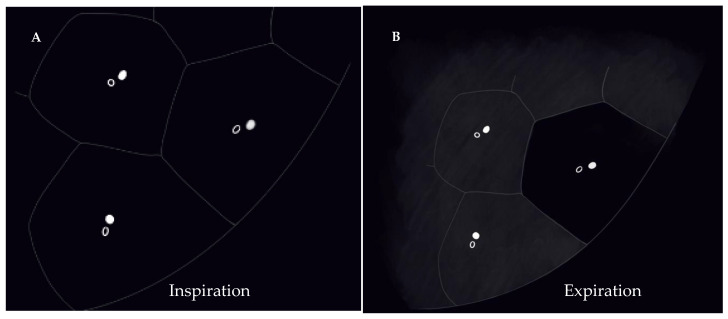
Air trapping (**A**,**B**). Axial inspiratory (**A**,**C**,**E**,**G**) and expiratory phase (**B**,**D**,**F**,**H**) CTs. On expiratory images the normal lung shows increase in the parenchymal density and decrease in volume. Interspersed geographic areas of air trapping lack the expected increase in density and volume reduction. Accentuated attenuation difference between areas of low and high density (32 versus 98 HU on image (**C**) and (**D**) respectively) indicates airway disease.

**Figure 5 diagnostics-12-02874-f005:**
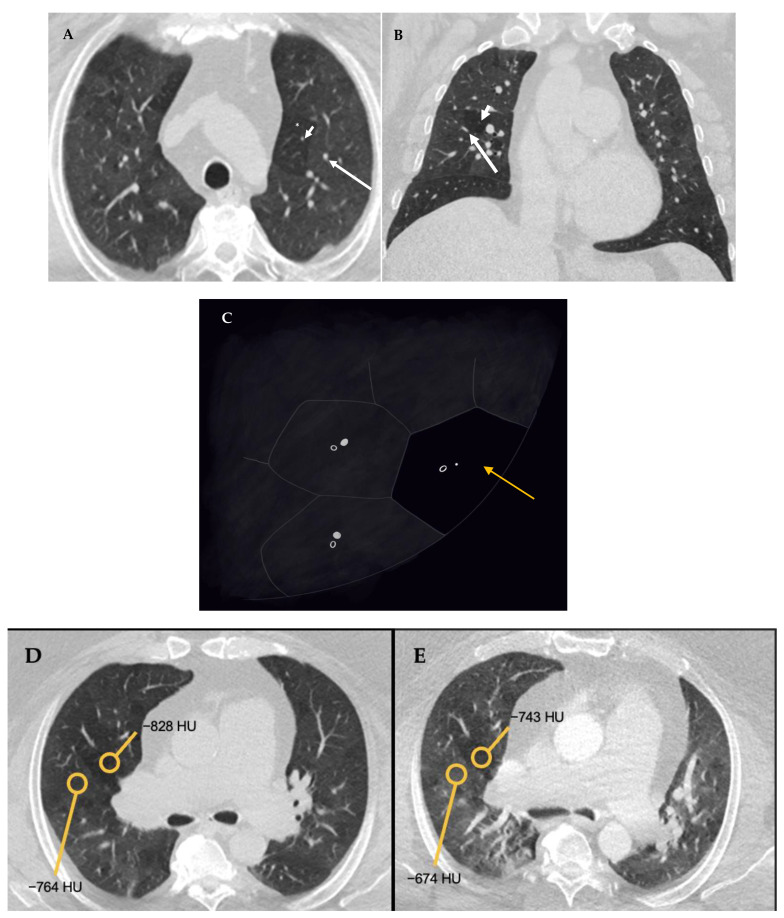
Mosaic perfusion (**C**). Axial (**A**) and coronal (**B**) inspiratory phase CTs show sharply demarcated regions of low attenuation (asterisk) interspersed in a background of normal (high) lung attenuation. Small caliber of vessels (short arrow) in the lucent areas relative to the normal lung vasculature (long arrow). Similar gradient of attenuation between low and high attenuation areas measuring 64 HU during inspiration (**D**) and 69 HU during expiration (**E**) indicating small vessel disease. Notice the expected decrease in volume of both hypo- and hyper-attenuating areas (**E**).

**Figure 6 diagnostics-12-02874-f006:**
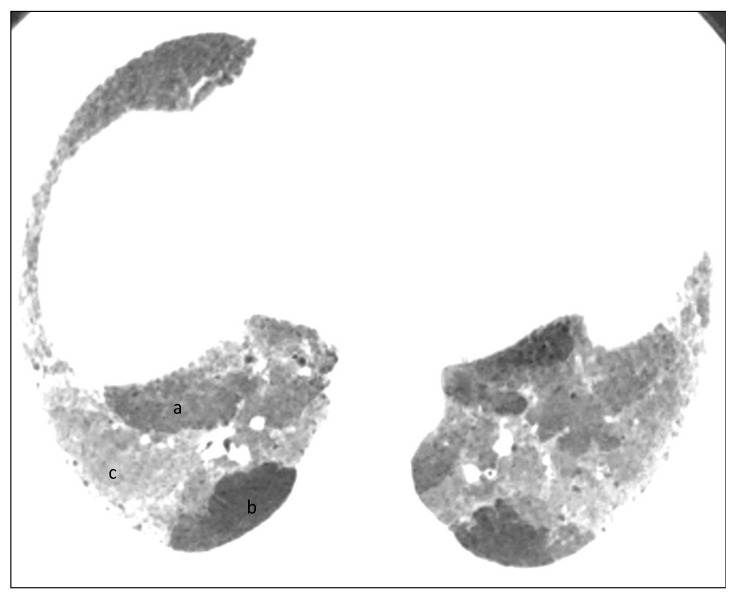
Axial inspiratory phase CT image shows sharply demarcated regions of three attenuations (three-density pattern): (**a**) Normal-appearing lung; (**b**) Lucent lung (i.e., regions of decreased attenuation and decreased vascularity; (**c**) High attenuation GGO.

**Figure 7 diagnostics-12-02874-f007:**
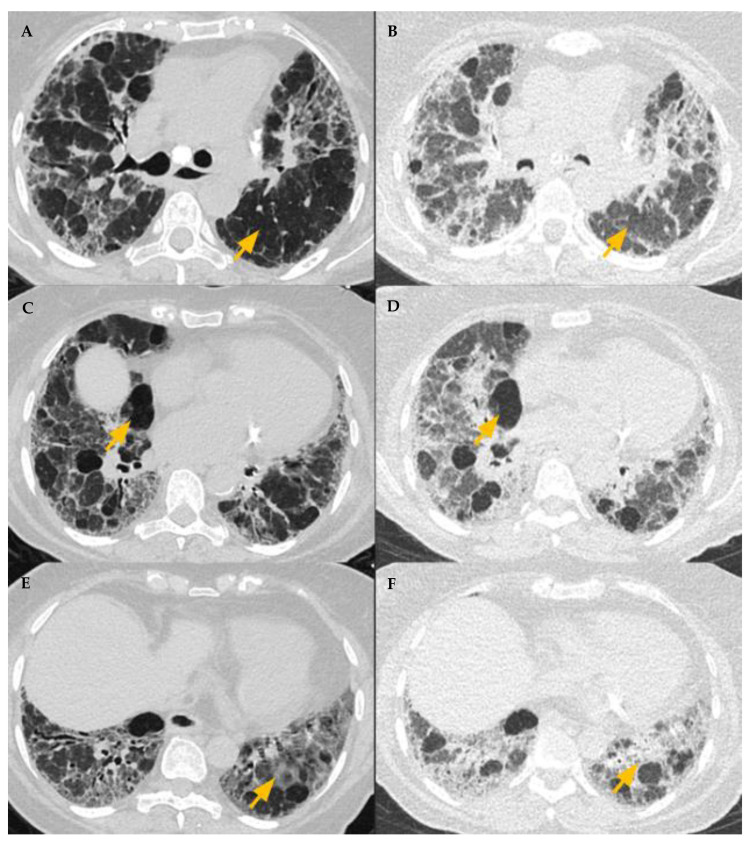
Three density pattern depicts simultaneous obstructive and infiltrative processes manifesting as air trapping (low density) and ground glass attenuation (high density), respectively, with areas of intervening normal lung parenchyma of intermediate density. Normal lung parenchyma (**A**) shows expected increased attenuation on expiration (**B**). Obstructive airway disease (air trapping) with decreased attenuation and vascularity on inspiration and expiration (**C**,**D**). GGO (**E**) with further increased attenuation on expiration (**F**).

**Figure 8 diagnostics-12-02874-f008:**
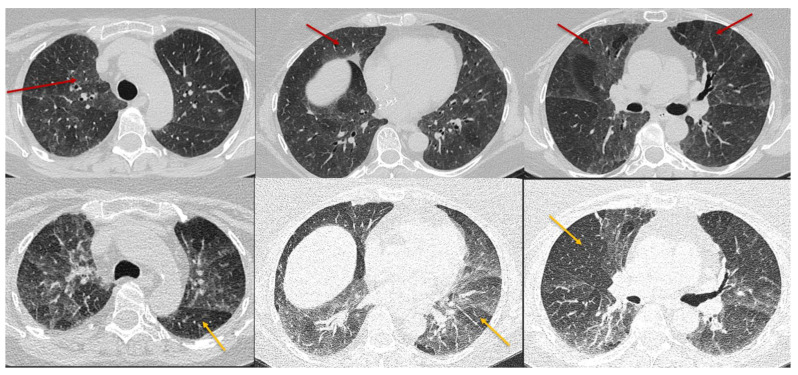
Typical non-fibrotic HP. Inspiratory phase CT (top row) shows ground glass opacities (red arrows) and expiratory phase CT (bottom row) shows air trapping (yellow arrows). Note the diffuse axial and craniocaudal distribution.

**Figure 9 diagnostics-12-02874-f009:**
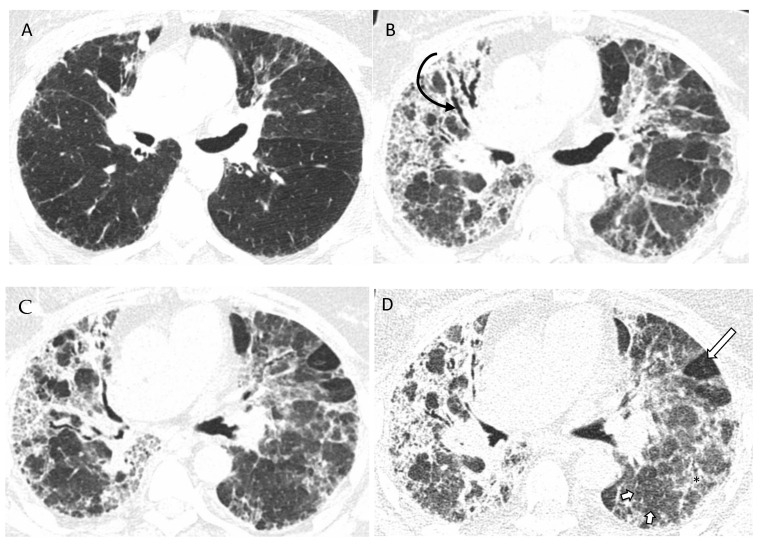
Typical fibrotic HP. Baseline CT (**A**) shows patchy GGOs. Follow up CT 3 years later (**B**–**D**) shows traction bronchiectasis (curved arrow), reticulations, patchy GGOs, and consolidations. Random axial and craniocaudal distribution of fibrosis. Axial inspiratory (**C**) and expiratory phase (**D**) shows three-density sign with expected increased attenuation of normal lung (short arrows) and GGOs (black asterisk). Lucent areas of decreased attenuation and vascularity depict air trapping (long arrow).

**Figure 10 diagnostics-12-02874-f010:**
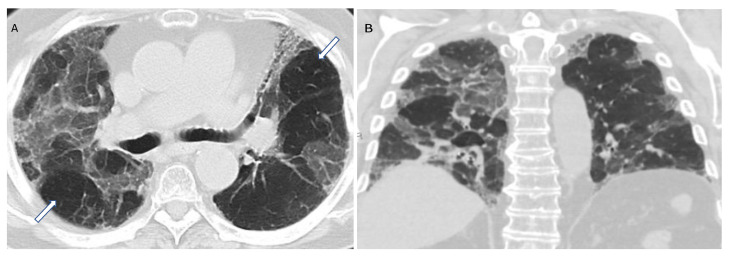
Compatible with fibrotic HP. Axial and coronal expiratory phase CT (**A**,**B**) shows coarse reticulations and minimal traction bronchiectasis superimposed on extensive upper lung predominant GGOs with peribronchovascular and subpleural distribution. Air trapping (arrows) is evident.

**Figure 11 diagnostics-12-02874-f011:**
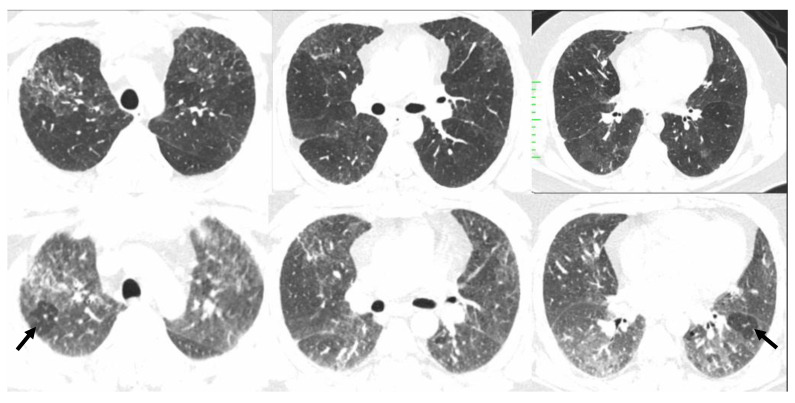
Compatible with fibrotic HP. Inspiratory phase CT (**top row**) shows ground glass opacities and subtle fibrosis. Note the variant upper lung predominant distribution. Expiratory phase CT (**bottom row**) shows air trapping (arrows).

**Figure 12 diagnostics-12-02874-f012:**
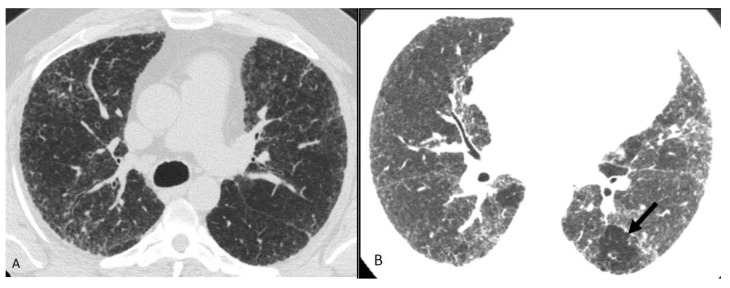
Compatible with fibrotic HP. Axial inspiratory phase CT shows (**A**) variant pattern of lung fibrosis with diffuse reticulations superimposed on a background of GGO and (**B**) lobular air trapping (arrow).

**Table 1 diagnostics-12-02874-t001:** Demographic and clinical features of hypersensitivity pneumonitis.

	Age	Unknown Antigen	VC, DLCO *	Lymphocytes (%) in BAL
Non-fibrotic HP	Younger	Less common	Low	Higher
Fibrotic HP	Older	More common	Lower	Lower

* DLCO = Diffusing capacity of carbon monoxide; VC = Vital capacity; BAL = Bronchoalveolar lavage.

**Table 3 diagnostics-12-02874-t003:** Patterns of non-fibrotic HP on chest HRCT scan.

Typical HP Pattern (Suggests a Diagnosis of HP)	Compatible with HP	Indeterminate for HP
At least one finding indicative of small airway disease		Not applicable
Air trapping		
Ill-defined <5 mm centrilobular nodules		
At least one finding indicative of parenchymal infiltration		
Mosaic attenuation	Diffuse and subtle GGO	
GGOs	Airspace consolidation	
	Lung cysts	
Distribution of findings		
Craniocaudal: diffuse +/− basal sparing	Craniocaudal: diffuse (variant: lower lobe predominance)	
Axial: diffuse	Axial: diffuse (variant: peribronchovascular)	

**Table 4 diagnostics-12-02874-t004:** Patterns of fibrotic HP on a chest HRCT scan.

Typical HP Pattern (Suggests a Diagnosis of HP)	Compatible with HP	Indeterminate for HP
At least one finding indicative of small airway disease	At least one finding indicative of small airway disease	Neither Typical nor Compatible with HP
Three-density pattern	Three-density pattern	HRCT Patterns: UIP pattern Probably UIP pattern Indeterminate for UIP Fibrotic NSIP pattern OP like pattern Truly indeterminate pattern
Air trapping	Air trapping	
Ill-defined <5 mm centrilobular nodules	Ill-defined <5 mm centrilobular nodules	
At least one finding indicative of pulmonary fibrosis	Variant pattern of fibrosis	
Coarse reticulations with distortion	UIP pattern of fibrosis	
Traction bronchiectasis	Extensive GGO and superimposed subtle fibrosis	
Honeycombing (not dominant)		
Distribution of findings	Variant distribution of fibrosis	
Random axially and craniocaudally	Craniocaudal: Upper lung zone predominant	
Mid zone predominant	Axial: peribronchovascular, subpleural	
Relative sparing of the bases		

**Table 5 diagnostics-12-02874-t005:** Histopathological features of HP.

HP	Probable HP	Indeterminate for HP
Nonfibrotic HP		
All three following features in at least one biopsy site	Both of the following features in at least one biopsy site	Presence of one of the following in at least one biopsy site
Cellular interstitial pneumonia	Cellular interstitial pneumonia	Cellular interstitial pneumonia
Cellular bronchiolitis	Cellular bronchiolitis	Cellular bronchiolitis
Poorly formed non-necrotizing granulomas		Selected IIP* patterns
Absence of features suggesting any alternative diagnosis in any biopsy site	Absence of features suggesting any alternative diagnosis in any biopsy site	Absence of features suggesting any alternative diagnosis in any biopsy site
Fibrotic HP		
All three following features in at least one biopsy site	Both of the following features in at least one biopsy site	Presence of the following in at least one biopsy site
Chronic fibrosing interstitial pneumonia	Chronic fibrosing interstitial pneumonia	Chronic fibrosing interstitial pneumonia
Airway-centered fibrosis	Airway-centered fibrosis	
Poorly formed non-necrotizing granulomas		
Absence of features suggesting any alternative diagnosis in any biopsy site †	Absence of features suggesting any alternative diagnosis in any biopsy site	Absence of features suggesting any alternative diagnosis in any biopsy site

† Features suggesting alternate diagnosis include four findings: plasma cells > lymphocytes, lymphoid hyperplasia, extensive well-formed sarcoidal with or without necrotizing granulomas, and aspirated particles.

**Table 6 diagnostics-12-02874-t006:** Algorithmic approach to diagnostic confidence for hypersensitivity pneumonitis based on exposure, BAL, histopathology, and imaging features.

HRCT						
	Typical for HP		Compatible with HP		Indeterminate for HP	
History of exposure and/or serum IgG testing	Exposure +	Exposure −	Exposure +	Exposure −	Exposure +	Exposure −
No BAL or BAL without lymphocytosis and either no or indeterminate histopathology	Moderate confidence	Low confidence	Low confidence	Not excluded	Not excluded	Not excluded
BAL lymphocytosis without histopathology sampling	High confidence	Moderate confidence	Moderate confidence	Low confidence	Low confidence	Not excluded
BAL lymphocytosis with indeterminate histopathology	Definite	High confidence	Moderate confidence	Moderate confidence	Low confidence	Not excluded
Probable HP histopathology	Definite	High confidence	High confidence	Moderate confidence	Moderate confidence	Low confidence
Typical HP histopathology	Definite	Definite	Definite	Definite	Definite	High confidence

## Data Availability

Not applicable.

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
