# Peer review of "Hypersensitivity Pneumonitis: A Pictorial Review Based on the New ATS/JRS/ALAT Clinical Practice Guideline for Radiologists and Pulmonologists"

_diagnostics, 2022, doi:10.3390/diagnostics12112874_

Round 1

Reviewer 1 Report

The authors have selected an important area for the review. The authors address the detailed description of radiologic findings for HP diagnosis. The manuscript was well-written and organized. However, there are some suggestions to improve the quality of the manuscript. 

1. Antigen identification is one of the most important parts of HP diagnosis. There are more causative antigens than the authors described, e.g., Mycobacterium spp. Further details have been described in the recent review based on the systematic review: "Exposure assessment in hypersensitivity pneumonitis: a comprehensive review and proposed screening questionnaire".

2. The statement "There are other nonspecific features, such as airspace consolidation and lung cysts, that can be found in HRCT of NFHP" in lines 154-155 is repetitive with the sentence in lines 150-151

3.The figures 3, 4A, 4B, and 5C are difficult to see. I would adjust the contrast of the figures. 

4. In figure 8, there is no yellow arrow, as the authors mentioned in the figure legend. 

5. Authors should additionally discuss doubtful situations. For example: 

      - Air trapping in IPF:  The previous study showed that approximately 40% of patients with the UIP pattern had air trapping. (Air trapping in usual interstitial pneumonia pattern at CT: prevalence and prognosis) How do we distinguish IPF from HP in these cases? 

       - Mosaic attenuation in patients with coexisting airway disease, e.g., in RA-ILD patients might have coexisting bronchiolitis obliterans or bronchiectasis, which CT findings can show mosaic attenuation or air trapping. 

I would add a discussion emphasizing the incorporation of clinical context and radiologic findings. Moreover, BAL lymphocytosis and histopathologic confirmation might be required in these cases.

Author Response

Point 1: Antigen identification is one of the most important parts of HP diagnosis. There are more causative antigens than the authors described, e.g., Mycobacterium spp. Further details have been described in the recent review based on the systematic review: "Exposure assessment in hypersensitivity pneumonitis: a comprehensive review and proposed screening questionnaire".

Response to point 1: Thank you very much for your comments. We have put other causative antigens in table 2 and cited the mentioned article as well.

Point 2: The statement "There are other nonspecific features, such as airspace consolidation and lung cysts, that can be found in HRCT of NFHP" in lines 154-155 is repetitive with the sentence in lines 150-151

Response to point 2: We have corrected the repetitive sentence.

Point 3: Figures 3, 4A, 4B, and 5C are difficult to see. I would adjust the contrast of the figures. 

Response to point 3: We have corrected the mentioned figures.

Point 4: In figure 8, there is no yellow arrow, as the authors mentioned in the figure legend. 

Response to point 4: We have corrected the arrows in the mentioned figure.

Point 5: 

Authors should additionally discuss doubtful situations. For example: 

      - Air trapping in IPF:  The previous study showed that approximately 40% of patients with the UIP pattern had air trapping. (Air trapping in usual interstitial pneumonia pattern at CT: prevalence and prognosis) How do we distinguish IPF from HP in these cases? 

       - Mosaic attenuation in patients with coexisting airway disease, e.g., in RA-ILD patients might have coexisting bronchiolitis obliterans or bronchiectasis, which CT findings can show mosaic attenuation or air trapping. 

Response to point 5 (1): Thank you for your suggestion. We added statements to section 6 to cover challenging scenarios and highlight the importance of clinical and lab data. We also changed the subheading of this section. We aimed to make this brief, yet informative to prevent disruption in the flow of the whole manuscript. 

-I would add a discussion emphasizing the incorporation of clinical context and radiologic findings. Moreover, BAL lymphocytosis and histopathologic confirmation might be required in these cases.

Response to point 5 (2): We added some statements to section 6 on the importance of clinical and lab data such as BAL in making the final diagnosis in ILD cases. Although this is very important, we believe more elaboration is beyond the scope of this pictorial review.

Also, it is worth mentioning that there were some wording issues with figure 1 and we corrected them.

Reviewer 2 Report

I congratulate you on writing this review, I’ve just a comment.

Comment 1. I believe the authors missed some opportunities to include a couple of interesting citations. These are enumerated below.

-                Page 2-3, Pathogenesis Paragraph, I think authors should add to citations also this paper [Fernández Pérez ER, Koelsch TL et al. Clinical Decision-Making in Hypersensitivity Pneumonitis: Diagnosis and Management. Semin Respir Crit Care Med. 2020 Apr;41(2):214-228. doi: 10.1055/s-0040-1701250. Epub 2020 Apr 12. PMID: 32279292].

-       Page 2-3, Pathogenesis Paragraph, I think authors should cite also this [Leone PM, Richeldi L. Current Diagnosis and Management of Hypersensitivity Pneumonitis. Tuberc Respir Dis (Seoul). 2020 Apr;83(2):122-131. doi: 10.4046/trd.2020.0012. Epub 2020 Mar 10. PMID: 32185914; PMCID: PMC7105432.]

Author Response

Point 1: Page 2-3, Pathogenesis Paragraph, I think authors should add to citations also this paper [Fernández Pérez ER, Koelsch TL et al. Clinical Decision-Making in Hypersensitivity Pneumonitis: Diagnosis and Management. Semin Respir Crit Care Med. 2020 Apr;41(2):214-228. doi: 10.1055/s-0040-1701250. Epub 2020 Apr 12. PMID: 32279292].

Response to point 1: Thank you very much for your comments. We have added the mentioned paper to the manuscript and the references.

Point2: Page 2-3, Pathogenesis Paragraph, I think authors should cite also this [Leone PM, Richeldi L. Current Diagnosis and Management of Hypersensitivity Pneumonitis. Tuberc Respir Dis (Seoul). 2020 Apr;83(2):122-131. doi: 10.4046/trd.2020.0012. Epub 2020 Mar 10. PMID: 32185914; PMCID: PMC7105432.]

Response to point 2: We have added the mentioned paper to the manuscript and the references.